# Effects of Insulin on Proliferation, Apoptosis, and Ferroptosis in Primordial Germ Cells via PI3K-AKT-mTOR Signaling Pathway

**DOI:** 10.3390/genes14101975

**Published:** 2023-10-22

**Authors:** Liu Ye, Xin Liu, Kai Jin, Yingjie Niu, Qisheng Zuo, Jiuzhou Song, Wei Han, Guohong Chen, Bichun Li

**Affiliations:** 1Key Laboratory of Animal Breeding Reproduction and Molecular Design for Jiangsu Province, College of Animal Science and Technology, Yangzhou University, Yangzhou 225009, China; xwl990422@163.com (L.Y.); xinliu6966@foxmail.com (X.L.); 007838@yzu.edu.cn (K.J.); 007510@yzu.edu.cn (Y.N.); 006664@yzu.edu.cn (Q.Z.); 2Institutes of Agricultural Science and Technology Development, Yangzhou University, Yangzhou 225009, China; 3Joint International Research Laboratory of Agriculture and Agri-Product Safety of Ministry of Education of China, Yangzhou University, Yangzhou 225009, China; 4Animal & Avian Sciences, University of Maryland, College Park, MA 20742, USA; songj88@umd.edu; 5Poultry Institute, Chinese Academy of Agricultural Sciences/Poultry Institute of Jiangsu, Yangzhou 225003, China; hanwei830@163.com; 6College of Biotechnology, Jiangsu University of Science and Technology, Zhenjiang 212100, China

**Keywords:** insulin, PGCs, proliferation, apoptosis, ferroptosis, PI3K-AKT-mTOR

## Abstract

Primordial germ cells (PGCs) are essential for the genetic modification, resource conservation, and recovery of endangered breeds in chickens and need to remain viable and proliferative in vitro. Therefore, there is an urgent need to elucidate the functions of the influencing factors and their regulatory mechanisms. In this study, PGCs collected from Rugao yellow chicken embryonic eggs at Day 5.5 were cultured in media containing 0, 5, 10, 20, 50, and 100 μg/mL insulin. The results showed that insulin regulates cell proliferation in PGCs in a dose-dependent way, with an optimal dose of 10 μg/mL. Insulin mediates the mRNA expression of cell cycle-, apoptosis-, and ferroptosis-related genes. Insulin at 50 μg/mL and 100 μg/mL slowed down the proliferation with elevated ion content and GSH/oxidized glutathione (GSSG) in PGCs compared to 10 μg/mL. In addition, insulin activates the PI3K/AKT/mTOR pathway dose dependently. Collectively, this study demonstrates that insulin reduces apoptosis and ferroptosis and enhances cell proliferation in a dose-dependent manner via the PI3K-AKT-mTOR signaling pathway in PGCs, providing a new addition to the theory of the regulatory role of the growth and proliferation of PGC in vitro cultures.

## 1. Introduction

Primordial germ cells (PGCs) are the progenitor cells of all animal germ cells, which eventually differentiate into sperm and eggs and are important carriers for the stable transmission of genetic information between generations [1,2,3] and have now become the tool cells to solve the bottleneck of avian biotechnology. Numerous experiments have demonstrated that exogenous PGCs (germinal crescent, blood, and gonads) from different time periods can colonize the gonads of homozygous recipient chickens and eventually develop into germline chimeras with the ability to spread germline [4,5,6]. In addition to the ability to proliferate in vitro, chicken primary germ cells can also be stored at low temperatures and genetically modified [7,8,9,10,11]. The results from these studies are not only a breakthrough for the establishment of avian genetic transformation technology but also a great contribution to the conservation and restoration of poultry germplasm resources. However, the number of PGCs isolated from embryos is far from sufficient to meet the actual needs due to the unclear genetic and molecular regulation mechanism of the growth and proliferation of chicken PGCs, which greatly limits the application of this technical system [12]. Studies have shown that the combined action of multiple growth factors and stromal cells alters the developmental program of PGCs by blocking their differentiation to mature germ cells while stimulating their proliferation in large numbers [13,14,15,16]. Pesce et al. reported that stem cell factor (SCF) and the leukemia inhibitory factor (LIF) promote the survival and proliferation of PGCs via the inhibition of apoptosis [17]. The addition of LIF, SCF, and basic fibroblast growth factor (bFGF) inhibits the apoptosis of 8.5 dpc PGCs cultured on feeder cells in vitro, promotes their proliferation and cell colony production, and maintains a pluripotent state [18,19,20,21]. Activin A also plays an important role in the PGC culture in vitro [22].

Insulin, a member of the peptide hormone family, has been reported to promote cell proliferation. Insulin enhances thyroid cell proliferation and tumor cell migration [23]. Insulin activates the phosphatidylinositol 3-kinase (PI3K)/protein kinase B (PKB, also known as Akt) pathway and mitogen-activated protein kinase (MAPK) signaling pathway to promote cell proliferation and apoptosis resistance [24]. In addition, insulin elevates cell viability. Studies have shown that mice administered with insulin exhibit higher follicle survival, lumen formation, and more mature oocyte production [25]. Insulin inhibits apoptosis after cleavage and the dissociation of cysteine aspartate and activates the insulin-like growth factor (IGF) receptor and PI3K/AKT cascade to promote cell survival [26]. Furthermore, insulin does exhibit various effects on cell behavior. Studies have shown that moderate amounts of insulin promote the proliferation of human arterial smooth muscle cells, while high concentrations of insulin reduce the efficiency of proliferation [27].

Lacking enough evidence from the literature, it is unclear whether the cellular activity response of PGCs undergoing different insulin doses may differ. Therefore, the objective of this study was to investigate the effects of insulin at different doses on the proliferation and cellular activity of PGCs.

## 2. Materials and Methods

### 2.1. Ethics Approval

All animal experiments were approved by the Institutional Animal Care and Use Committee of Yangzhou University (approval number: 132-2022). All procedures were strictly performed in accordance with the Regulations for the Administration of Affairs Concerning Experimental Animals (Yangzhou University, Yangzhou, China, 2012) and the Standards for the Administration of Experimental Practices (Yangzhou, China, 2008).

### 2.2. Isolation, Culture, and Identification of PGCs

The fertilized breeding eggs of Rugao yellow chickens were obtained from the Poultry Live Preservation Gene Bank, Institute of Poultry Research, Chinese Academy of Agricultural Sciences (Yangzhou, China). Fresh 4.5 d fertilized embryonic eggs were taken and washed in water with Neosporin disinfectant and 75% alcohol. Under aseptic conditions, the egg shell was gently tapped with the blunt end of forceps, and the tipped part of the forceps was used to peel the cracked egg shell. Then, the embryo was gently picked out, and the embryo was cleaned with PBS two times. Then, the internal organs were removed under a body microscope, and the germinal crests of the embryos were taken from the anterior 1/3 dorsal spine of the hindgut and put into 1.5 mL centrifugation tubes, which were then centrifugated at 370× *g* for 6 min, and the supernatant was removed. The supernatant was washed again with PBS, added to 100 μL trypsin (0.25%) digestion, ground for 1 min, added 200 µL PGCs complete medium (Table 1) to terminate the digestion, centrifuged at 370× *g* for 3 min, discarded the supernatant, resuspended to a 48-well plate, and cultured at 37 °C, 5% CO_2_ incubator. Mulberry-like PGC cells could be observed under the inverted microscope. In our experiment, we had a nutritional deficiency of insulin. PGCs were seeded in 24-well plates and cultured in complete media without insulin B27 was replaced by B27 minus insulin (Gibco-A1895601, Shanghai, China) supplemented with gradient concentrations of insulin at 0, 5, 10, 20, 50, and 100 μg/mL, for 72 h.

Cells were collected in well plates, rinsed twice with phosphate-buffered saline (PBS), then drop the cells onto the slide, fixed with 4% paraformaldehyde for 30 min, rinsed three times with PBS; 0.1% Triton permeable membrane was treated for 15 min, rinsed three times with PBS, PBS with 10% FBS added was closed for 2 h against light and then a primary antibody SSEA-1 (Abcam-ab16285, 1:400, Shanghai, China) was added dropwise and incubated at 37 °C for 2 h and then overnight at 4 °C; PBS washed with a primary antibody, a secondary antibody (BOSTER-BA1101, 1:400, Wuhan, China) was added dropwise, incubated at 37 °C. After incubation for 2 h at 37 °C, and then the secondary antibody was washed with PBS, stained with 5 ng/μL of Hoechst (Ribobio-U1202, Guangzhou, China) for 15 min, and washed the stain, buffer glycerol (50% glycerol, 50% PBS) to seal the film after dropping the oil. The fluorescence was observed under the microscope.

### 2.3. PGCs Proliferative Capacity Test

The proliferative capacity was detected using CCK-8 kit (A311-01, Vazyme, Nanjing, China) according to the manufacturer’s instruction. Briefly, 100 μL cell solution (1 × 10^6^/mL) of PGCs at logarithmic growth stage was seeded in 96-well plates in triplicate. Then, 10 μL of CCK-8 solution was added, followed by the incubation at 37 °C and 5% CO_2_ for 2 h. The absorbance at 450 nm was read using microplate reader (TECAN SPARK, Shanghai, China).

After the Edu-labeling (C10310-1, Ribobio, Guangzhou, China), fixation, and permeabilization, PGCs were incubated with Click reaction solution for 30 min at room temperature in dark and then stained with Hoechst33342 for 10 min at room temperature in dark. Fluorescence was observed under fluorescence microscope (LEICA DMi8, Beijing, China).

### 2.4. Quantitative PCR

Invitrogen TRIzol (ThermoFisher-15596026, Shanghai, China) and FastKing gDNA Dispelling RT SuperMix (TIANGEN-KR118, Beijing, China) were used for RNA extraction and cDNA synthesis. 1 μg RNA was used for cDNA synthesis. The kit method extracted RNA, and reverse transcribed cDNA was stored frozen at −20 °C. Primers for ferroptosis, apoptosis, PI3K-Akt-mTOR, and cell cycle-related genes were designed according to GenBank using BLAST software in NCBI (Table 2), and ACTIN was used as an internal reference gene. The qRT-PCR technique was used to detect the mRNA expression levels of each gene in PGCs after the treatment of insulin at different concentrations. Moreover, 10 μL qRT-PCR reaction system consists of 5 μL 2× Universal SYBR Green Fast qPCR Mix, 0.4 μL upstream primer (10 μM), 0.4 μL downstream primer (10 μM), 2 μL cDNA, ddH_2_O to 10 μL. Reaction procedure: 95 °C for 3 min; 95 °C for 5 s; and 60 °C for 30 s with 40 cycles in total. Three biological replicates were set for the culture of PGCs, and three technical replicates were set for each sample. Expression levels were quantified using the 2^−ΔΔCT^ method.

### 2.5. Western Blot Test

PGCs were collected and lysed using Radio-Immunoprecipitation Assay (RIPA) lysis solution (Beyotime-P0013B, Shanghai, China). Extracted total proteins were separated and transferred to PVDF membrane (Beyotime-FFP39, Shanghai, China) via electrophoresis. Blots were blocked using 5% skimmed milk at room temperature for 2 h, followed by the incubation of primary antibody solution (PI3K (BIOSS-bsm-33219M, Beijing, China), p-PI3K (BIOSS-bs-6417R, Beijing, China), AKT (Affinity-AF6261, Liyang, China), p-AKT (BIOSS-bs-0876R, Beijing, China), GAPDH (Proteintech-10494-1-AP, Wuhan, China)) at 4 °C overnight. After 3 washes with TBST, the blots were incubated with secondary antibody solution (Goat Anti-Mouse IgG (CWBIO-CW0102, Taizhou, China), Goat Anti-Rabbit IgG (CWBIO-CW0103, Taizhou, China) at room temperature for 2 h. The blots were washed 3 times with TBST (Solarbio-T1082, Beijing, China) and subjected to imaging using chemiluminescent substrate.

### 2.6. PGCs Ferrous Ion Detection

The cytosolic ferrous colorimetric assay kit (E-BC-K881-M, Elabscience, Wuhan, China) was used to detect ferrous ions according to the manufacturer’s instructions. Briefly, 1 × 10^6^/mL PGCs were seeded in 96-well plates and incubated with staining solution at 37 °C for 10 min. The absorbance at 593 nm was read by a microplate reader (TECAN SPARK, Shanghai, China).

### 2.7. PGCs Oxidation Levels Detection

PGCs were inoculated in 24-well plates at 1 × 10^5^/mL. Different concentrations of insulin were added and incubated for 72 h. The cell precipitates were collected.

The Malondialdehyde (MDA) content was detected according to MDA kit (KGT004-1, KeyGEN, Nanjing, China). The sample was resuspended in PBS and lysed via low-temperature ultrasonication, and the protein concentration was determined using bicinchoninic acid (BCA) method, and the MDA working solution was configured. Then, 100 µL of sample was taken, mixed with 1 mL of working solution, heated at 95 °C for 1 h, cooled to room temperature, and centrifuged at 6000× *g* for 10 min, and the supernatant was taken. Next, 200 µL of sample was added into a 96-well plate, and the absorbance at 532 nm was detected using enzyme marker (TECAN SPARK, Shanghai, China).

The GSH/GSSG content was detected according to GSH and GSSG kit (S0053, Beyotime, Shanghai, China). PGCs in logarithmic phase were inoculated into 6-well plates at 3 × 10^5^ cells/well, and protein remover S solution was added at 3 times the volume of cell sedimentation when the cell fusion was about 50%. Cells were rapidly frozen and thawed 3 times and centrifuged at 12,000× *g* for 10 min, and the supernatant was used for GSH assay. The absorbance at 412 nm was detected using enzyme marker (TECAN SPARK, Shanghai, China), and the GSH content was calculated from the standard curve.

### 2.8. PGCs Apoptosis Detection

PGCs were inoculated in 6-well plates at a density of 1 × 10^6^/mL (1 mL per well) and incubated with different concentrations of insulin for 72 h. The cell sediment was collected. The cells were washed twice with PBS and double-stained with AnnexinV/PI (YEASEN-40302ES, Shanghai, China), incubated for 5 min at room temperature with no light, and then detected on the machine.

### 2.9. Statistical Analysis

Statistical analyses of data were performed using one-way ANOVA of SPSS 22.0 and GraphPad Prism 8 software (GraphPad Software Inc., San Diego, CA, USA) and presented as the mean ± SEM. *p* < 0.05 was considered to represent a statistically significant difference.

## 3. Results

### 3.1. Insulin Regulates PGC Proliferation in a Dose-Dependent Manner

SSEA-1 was specifically expressed on chicken PGCs, and via immunofluorescence, we confirmed that the isolated cells were PGCs (Figure 1A). Studies have shown that insulin is an essential factor for the cell culture. B-27 is an important factor in culturing PGCs in our laboratory, which contain insulin, and the dose of insulin is not yet known. To investigate the effect of insulin on the culture of PGCs, the insulin was removed from the culture medium, which led to abnormal cell morphology of PGCs (Figure 1B,C): The proliferation of PGCs was decreased with an irregular shape of cells after removal of insulin, while it was rescued after the addition of insulin back to the culture medium. These results indicate that insulin is indispensable in the process of culturing PGCs. In order to screen the optimal dosage of insulin for maintaining PGCs, we cultured PGCs in a medium containing different concentrations of insulin and detected the cellular activity of PGCs.

PGCs exposed in a low concentration range (10 μg/mL) exhibited a good state, while those in higher concentrations of insulin presented obvious clumps and cell fragments (Figure 2A). Insulin significantly promoted the proliferation of PGCs in the low concentration range (10 μg/mL) (*p* < 0.05). Interestingly, insulin at higher concentrations had less effect on the proliferative rate of PGCs (*p* < 0.05) (Figure 2B,C). These results tentatively imply that insulin regulates the proliferative capacity of PGCs dose dependently.

To further verify the effect of insulin on the proliferative capacity of PGCs, we examined the expression levels of cell cycle-related genes (CCND1, ABL1, CCNB1, and CCNF) in PGCs after insulin treatment at different concentrations. The expression of cell cycle-related genes was significantly increased by insulin at 10 μg/mL with less elevation when cells were exposed to higher doses of insulin (*p* < 0.05) (Figure 3A–D), which was consistent with the results related to cell proliferation assay. These findings suggest that insulin is essential for the culture of PGCs, and insulin mediates PGC proliferation in a typical dose-dependent manner with an optimal concentration of 10 μg/mL.

### 3.2. Insulin Can Inhibit the Apoptosis Level of PGCs and Improve the Cell Viability

Studies have shown that insulin can participate in the regulatory process of apoptosis. In the study of insulin affecting the proliferation ability of PGCs, we found that insulin has a greater effect on the cell viability of PGCs. To explore the factors of insulin affecting the viability of PGCs, we examined the level of apoptosis during the culture of PGCs with different concentrations of insulin. The results showed the following (Figure 4A–E): BCL-2 expression was significantly upregulated (*p* < 0.05); C-myc, Bax, caspase3, and caspase9 expression was significantly downregulated (*p* < 0.05) after the addition of insulin. It is well known that BCL-2 and C-myc, Bax, caspase3, caspase9 are marker genes for apoptosis levels, and significant apoptosis occurs in cells with downregulation of BCL-2 levels and upregulation of C-myc, Bax, caspase3, caspase9 levels. In order to further clarify the effect of insulin on the apoptosis level of PGCs, we performed flow cytometry analysis via PI. The results showed (Figure 4F,G) that, consistent with the quantitative results, the apoptosis level of PGCs is significantly inhibited via the addition of insulin (*p* < 0.05). Thus, our results suggest that the addition of insulin reduces the apoptosis of PGCs.

### 3.3. Insulin Can Inhibit the Ferroptosis Level of PGCs and Improve Cell Viability

Ferroptosis is a specific form of cell death that has a significant effect on cell viability. To investigate the effect of ferroptosis levels in PGCs cultured with different concentrations of insulin, we used qRT-PCR to detect the expression of ferroptosis-related genes (GPX4, SLC7A11, and NOX4). The results showed the following (Figure 5A–C): the expression of GPX4 and SLC7A11, marker genes as negative regulators of cellular ferroptosis, showed a trend of upregulation (10 μg/mL) and then downregulation with the increase in insulin addition level during the culture of PGCs, while NOX4, a marker gene for positive regulation, showed the exact opposite trend (*p* < 0.05). These results tentatively suggest that insulin has the ability to inhibit the level of ferroptosis in PGCs.

To further explore the effect of insulin on the ferroptosis level of PGCs, the ferrous ion content in PGCs was examined in this experiment, and the results showed the following (Figure 5D): Similar to the results of the ferroptosis marker gene, with the increase in insulin addition level, the ferrous ion content in PGCs showed a trend of first decreasing and then upregulating (*p* < 0.05), and the insulin addition concentration of 10 μg/mL, the ferrous ion content was the lowest, and the intracellular ferrous ion content showed a positive correlation trend with the level of cellular ferroptosis. Therefore, our results suggest that insulin can inhibit the level of ferroptosis in PGCs to improve cell viability.

We also examined the redox levels and the degree of cellular damage in PGCs, and the results showed the following (Figure 6A–D): with the increase in insulin addition level, the total GSH level in PGCs did not change significantly, the oxidized GSSG level showed a trend of first decreasing and then upregulating, the reduced GSH level showed a trend of first upregulating and then decreasing (*p* < 0.05), and MDA showed a trend of first decreasing and then upregulating trend. At the insulin concentration of 10 μg/mL, PGCs showed the highest reduced GSH, the lowest oxidized GSSG, and the lowest MDA level, while the cellular oxidation level and the degree of cellular damage showed a positive correlation with cellular ferroptosis. In conclusion, insulin was able to inhibit the ferroptosis level of PGCs to improve cell viability.

### 3.4. Insulin Can Affect Cell Viability of PGCs via PI3K-Akt-mTOR Signaling Pathway

It is well known that PI3K-Akt-mTOR signaling is an important signaling pathway that regulates cell viability. So, does insulin regulate cell viability through this signaling pathway? To prove this hypothesis, we detected the expression of PI3K-Akt-mTOR signaling pathway-related proteins via Western Blot, and the results showed the following (Figure 7A–C): p-PI3K/PI3K protein expression increased after insulin addition and protein expression was upregulated with increasing concentration; p-Akt/Akt protein expression was first upregulated and then downregulated, and 10 μg/mL was the highest (*p* < 0.05). The results of the qRT-PCR assay showed (Figure 7D–F) that the expression levels of *PI3K*, *Akt*, and *mTOR* genes were significantly increased after the addition of insulin, and the highest gene expression levels were observed at 10 μg/mL insulin (*p* < 0.05). These results show that insulin activates the PI3K-Akt-mTOR signaling pathway and that insulin concentration determines the pathway activation capacity.

## 4. Discussion

Insulin can bind to insulin receptors on cell membranes to regulate various metabolic pathways in cells, increase the synthesis of fatty acids and glucose, and play an important role in cell growth. In vitro studies have shown that insulin has a significant mitogenic effect on cell proliferation [28,29]. Jemima et al. found that PGCs do not proliferate if insulin is removed from B-27 supplementation [30]. The addition of insulin to the culture medium of PGCs better maintains survival and promotes an increase in the number of PGC colonies [31]. In this present study, we found that with the removal of insulin from B-27, PGCs significantly did not proliferate, and the addition of insulin significantly promoted the proliferation of PGCs and increased the expression of cell cycle genes during all phases of mitosis, in agreement with previous studies. In addition, it was shown that insulin promotes cardiomyocyte proliferation and survival via the activation of PI3K/Akt [32,33,34,35]. Phosphorylation levels of the AKT signaling pathway and its downstream proteins may play a key role in maintaining the undifferentiated state of stem cells and regulating stem cell self-renewal and proliferation [36]. mTOR is one of the downstream targets of the PI3K/Akt signaling pathway. It has been shown that mTOR inhibits excessive autophagy and stimulates cell growth, proliferation, and survival as well as metabolism via the upregulation of protein and lipid synthesis [37,38]. We confirmed that insulin addition positively affected the PI3K-Akt-mTOR signaling pathway and promoted the proliferation of PGCs, and the proliferation efficiency of PGCs showed a positive correlation trend with the expression of genes and proteins related to this signaling pathway, which is consistent with the above study. This study elucidated the effects of insulin on PGCs from different concentrations and found that high concentrations of insulin inhibited the proliferation efficiency of PGCs, complementing the theoretical basis for the study of insulin additive doses in PGCs in vitro culture.

Insulin has been shown to protect hepatocytes from saturated fatty acid-induced apoptosis by inhibiting c-Jun NH2-terminal kinase activity [39]. Cells die if Bax homodimers predominate, while they survive when Bcl-2 and Bax heterodimers predominate [40]. The overexpression of C-myc is accompanied by apoptosis [41]. Caspase3 and caspase9 are key enzymes in the apoptotic process and play a critical role in apoptosis [42,43]. We found that the addition of insulin inhibited the expression of pro-apoptotic genes Bax, Caspase-3, Caspase-9, and C-myc and promoted the expression of apoptosis-suppressing gene Bcl-2 in PGCs, while the inhibition of apoptosis in PGCs via insulin was confirmed using flow cytometry, which was consistent with the previous results. In addition, it was shown that the inhibition of the PI3K-AKT-mTOR signaling pathway inhibits insulin release and promotes apoptosis [44]. Akt can deliver anti-apoptotic or survival signals by stimulating antioxidant response element activity and subsequent antioxidant enzyme expression and by inducing the upregulation of heme oxygenase expression via the activation of transcription factor e2-related factor 2 [45,46]. We found that the addition of insulin promoted the expression of PI3K-AKT-mTOR signaling pathway-related genes and proteins, confirming the important role of the PI3K-AKT-mTOR signaling pathway on the apoptosis of PGCs.

In addition, it was shown that insulin protects cardiomyocytes from ferroptosis by activating the PI3K/Akt pathway, reducing reactive oxygen species, and maintaining mitochondrial transmembrane potential [47]. The expression of GPX4 and SLC7A11 is reduced when ferroptosis occurs in cells, while NOX4 expression is upregulated [48,49], which is consistent with our study. Meanwhile, it was shown that MaR1 inhibited ferroptosis in mouse hepatocytes and reduced the levels of malondialdehyde (MDA), reduced glutathione (GSH), GSH/oxidized glutathione (GSSG), and iron content [50]. We found that the addition of appropriate insulin reduced intracellular levels of ferric ions, oxidized GSSG, reduced GSH and MDA, and protected PGCs from ferroptosis, in agreement with previous results [50]. In addition, hyperinsulinemia can upregulate hepatic transferrin receptor 1 (TFR1) via PI3K/AKT/mTOR/IRP2 pathway, causing iron overload damage in rat liver, resulting in iron death and reduced cell viability [51], and we found that whereas high insulin concentration leads to ferroptosis in PGCs, high insulin concentration also activates this signaling pathway, validating the results of previous experiments. Overall, we found that insulin can affect ferroptosis levels in PGCs via the PI3K-Akt-mTOR signaling pathway.

## 5. Conclusions

Collectively, insulin was essential for the in vitro culture of PGCs. The proliferation ability and signaling pathway activation of PGCs were the strongest as well as the least affected by apoptosis and iron death when insulin was added at a concentration of 10 μg/mL. This study provides a valuable reference for the development of a stable and effective in vitro culture medium for PGCs.

## Figures and Tables

**Figure 1 genes-14-01975-f001:**
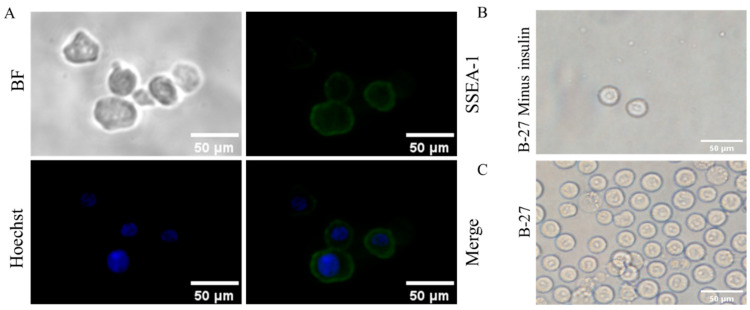
Morphological observation and identification of PGCs. (**A**) Identification of isolated PGCs. (**B**) Cell morphology of PGCs cultured with B-27 (minus insulin). (**C**) Cell morphology of PGCs cultured with B-27.

**Figure 2 genes-14-01975-f002:**
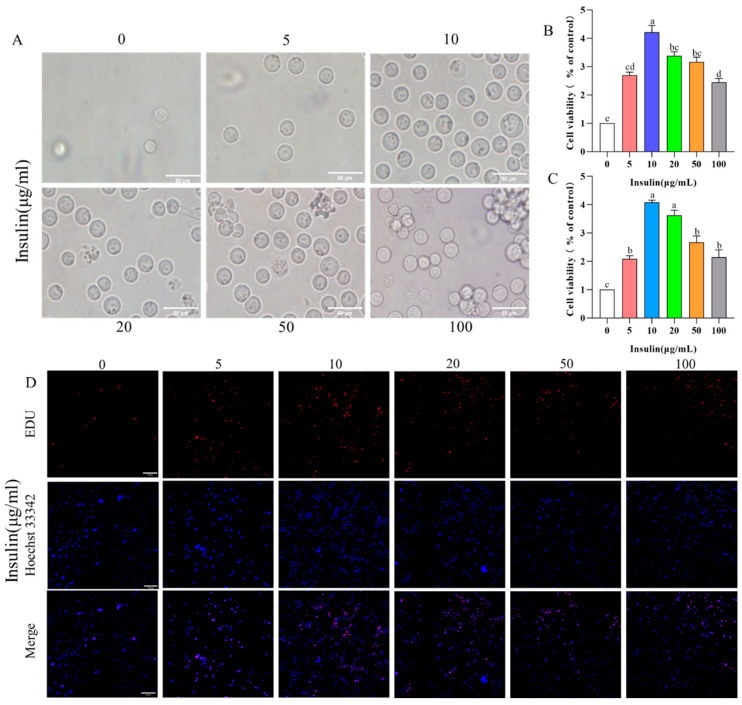
Insulin mediates PGC growth in a dose-dependent way. (**A**) Cell morphology of PGCs under insulin treatment at different concentrations. (**B**,**C**) Cell proliferation rate of PGCs using CCK-8 kit (**B**) and EDU kit (**C**). (**D**) EDU staining of PGCs. ^a,b,c,d,e^ Means within a row with no common superscript differ significantly (*p* < 0.05).

**Figure 3 genes-14-01975-f003:**
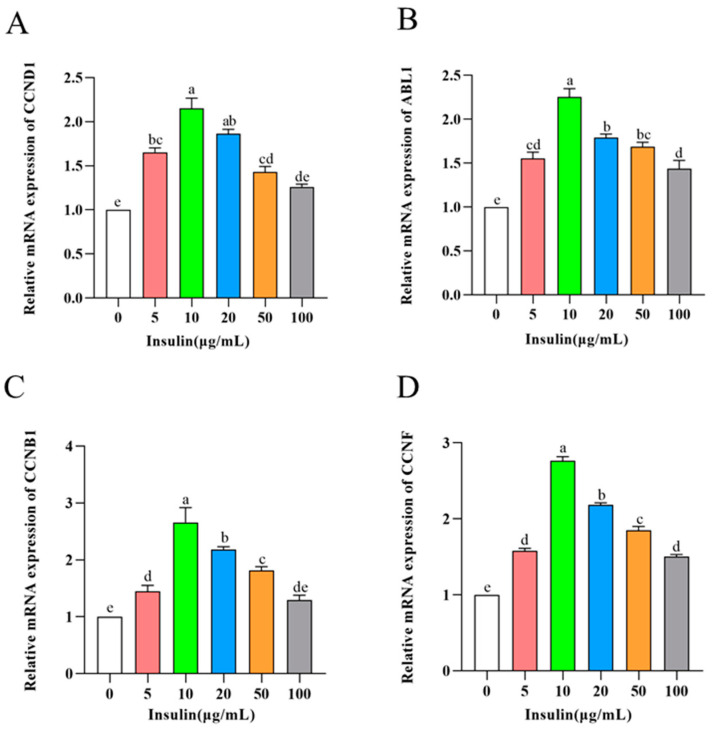
mRNA levels of *CCND1* (**A**), *ABL1* (**B**), *CCNB1* (**C**), *CCNF* (**D**). ^a,b,c,d,e^ Means within a row with no common superscript differ significantly (*p* < 0.05).

**Figure 4 genes-14-01975-f004:**
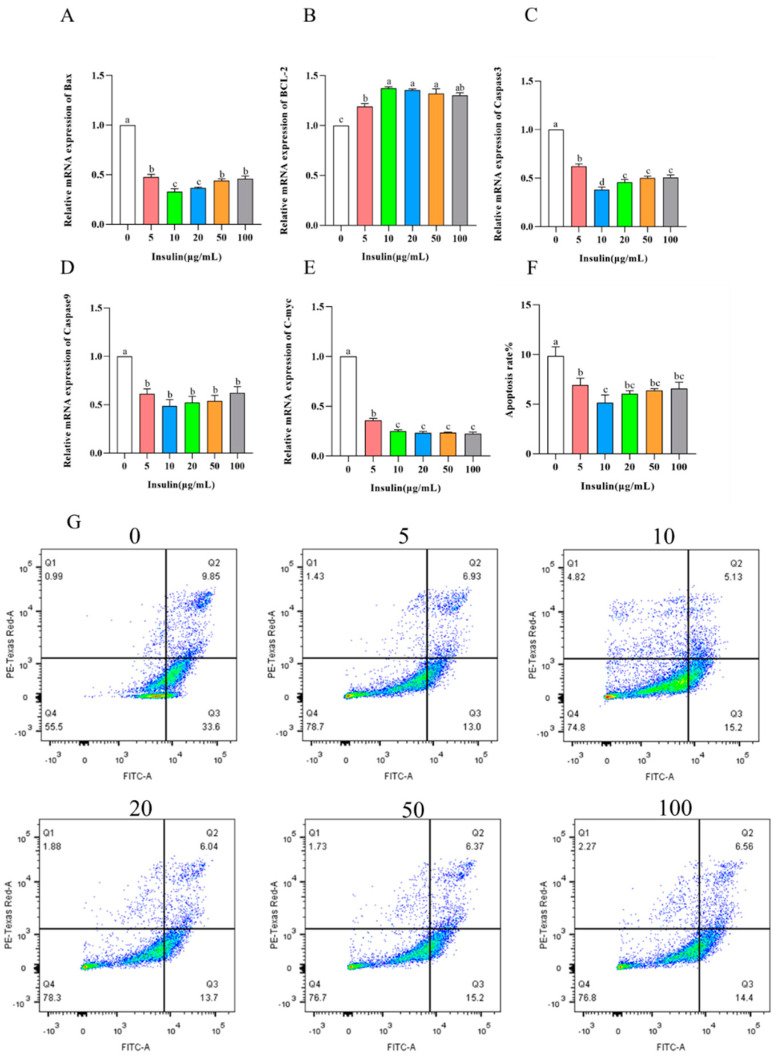
Insulin reduces cell apoptosis in PGCs. (**A**–**E**) mRNA levels of apoptosis-related genes, *BAX* (**A**), *BCL2* (**B**), *CASP3* (**C**), *CASP9* (**D**), *C-myc* (**E**). (**F**,**G**) Apoptosis rate (**F**) and flow cytometric analysis (**G**) of PGCs under insulin treatment. ^a,b,c,d^ Means within a row with no common superscript differ significantly (*p* < 0.05).

**Figure 5 genes-14-01975-f005:**
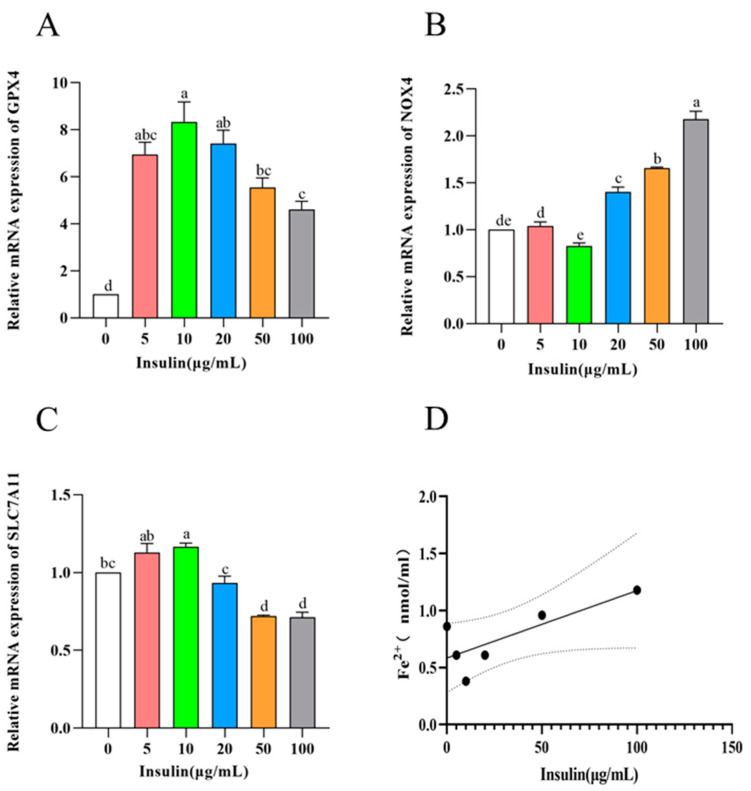
Insulin reduces ferroptosis in PGCs. (**A**–**C**) mRNA levels of ferroptosis-related genes *GPX4* (**A**), *NOX4* (**B**), and *SLC7A11* (**C**). (**D**) Iron content in PGCs. ^a,b,c,d,e^ Means within a row with no common superscript differ significantly (*p* < 0.05).

**Figure 6 genes-14-01975-f006:**
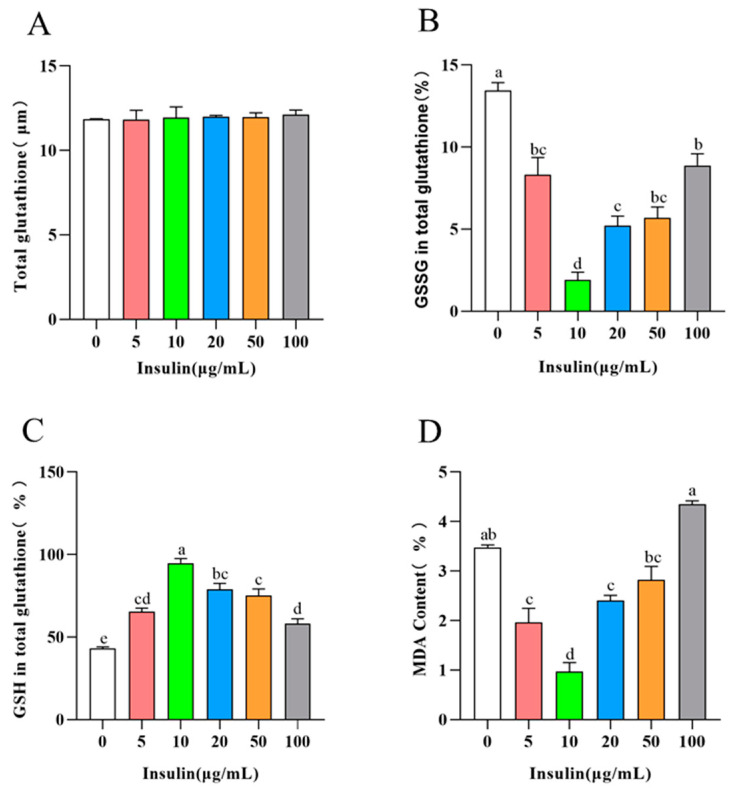
Insulin reduces the redox levels and the degree of cellular damage in PGCs. (**A**) Total glutathione in PGCs. (**B**) Oxidative GSSG in PGCs. (**C**) GSH in total glutathione in PGCs. (**D**) MDA level in PGCs. ^a,b,c,d,e^ Means within a row with no common superscript differ significantly (*p* < 0.05).

**Figure 7 genes-14-01975-f007:**
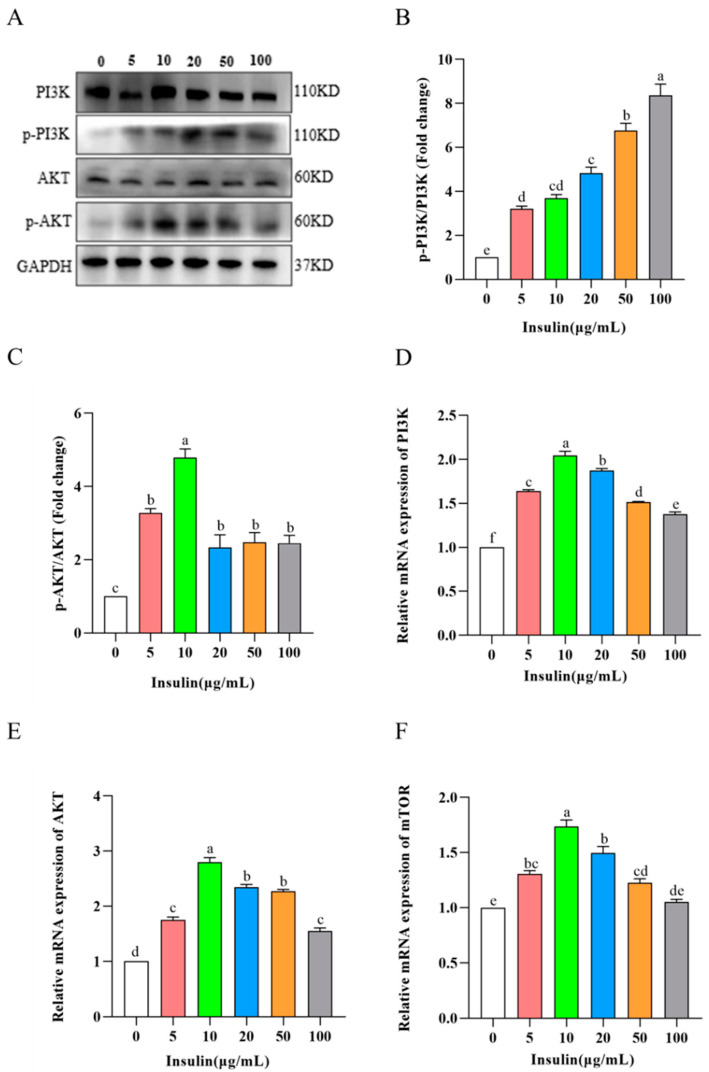
Insulin activates PI3K/AKT/mTOR pathway. (**A**–**C**) Western blot and quantification of p-PI3K/PI3K (**A**,**B**) and p-Akt/Akt (**A**,**C**). (**D**–**F**) mRNA levels of *PI3K*, *AKT1*, and *MTOR*. ^a,b,c,d,e,f^ Means within a row with no common superscript differ significantly (*p* < 0.05).

**Table 1 genes-14-01975-t001:** Preparation of PGCs complete medium.

KO-DMEM Basal Medium			
**Composition**	**Source ***	**Concentration**	**Volume**
DMEM	Meilunbio-PWL037	75%	37.5 mL
Ultra-filtered Water	Sigma-W3500	24%	12 mL
CaCl_2_·2H_2_O	Sigma-C7902	0.15 mM	0.5 mL
**PGCs complete medium**			
**Composition**	**Source**	**Concentration**	**Volume**
KO-DMEM basal medium			46.632 mL
B-27^TM^ supplement	Gibco-17504044	1×	1 mL
GlutaMax	Gibco-35050061	2 mM	0.5 mL
MEM NEAA	Gibco-11140050	1×	0.5 mL
2-Mercaptoethanol	Gibco-21985023	0.1 mM	91 µL
Chicken serum	Gibco-16110082	0.2%	100 µL
EmbryoMax Nucleosides	Sigma-ES-008-D	1×	0.5 mL
Sodium pyruvate	Gibco-11360070	1.2 mM	0.6 mL
Ovalbumln	Sigma-A5503	0.2%	0.1 g
Sodium heparin	MCE-HY-17567A	0.01%	50 µL
Basic fibroblast growth factor	MCE-HY-P70600	1×	20 µL
Human Activin A	MCE-HY-P70311	1×	25 µL
Pen Strep	Gibco-15070063	1×	0.5 mL

*: Meilunbio (Dalian, China), Sigma (Shanghai, China), Gibco (Shanghai, China), MCE (Shanghai, China).

**Table 2 genes-14-01975-t002:** The primer sequences for qRT-PCR.

Primer Name	Primer Sequence (5′-3′)
GPX4	F: CCTACCTGTAGAGCTGCGTG
	R: AATCTTCGGGTCTGCCTCAC
NOX4	F: CTGGCTCTCACTGAACGTGT
	R: GCTAACACACAATCCTAGCCCT
SLC7A11	F: CAGCCTCCTGAATTTCCTCAGT
	R: AGGCACCTTGAAAGGACGAG
CCND1	F: TTTGTTCGGCTCGAAGAGAGC
	R: TCATCGCCAAGGGGAAAACT
ABL1	F: AGCTGCCGCTGCTCC
	R: TTAGCGAAGGCCAAAGCAAC
CCNB1	F: CTGCTTTCCGTACCAATGGC
	R: CAGTCCGTTTTCTTGGGCAC
CCNF	F: GGAGTCAGCTTCAGCCTCAG
	R: AGAAGCATCTGGAACAGCGG
C-MYC	F: ACACAACTACGCTGCTCCTC
	R: TTCGCCTCTTGTCGTTCTCC
BCL-2	F: CCAAGCAAAAAGAGGAGTCACG
	R: ACCGTTATACCTAATGCAGCCA
BAX	F: TCCATTCAGGTTCTCTTGACC
	R: GCCAAACATCCAAACACAGA
CASPASE3	F: CTGAAGGCTCCTGGTTTA
	R: TGCCACTCTGCGATTTAC
CASPASE9	F: ATTCCTTTCCAGGCTCCATC
	R: CACTCACCTTGTCCCTCCAG
PI3K	F: CTTCTGGAGTCCTATTGTCG
	R: CACCTTCTGGGTCTCATCTT
AKT	F: GCCGTGAGCCCAGTTAGG
	R: AGCTACTTATGGCTGCGGGA
MTOR	F: AACCACTGCTCGCCACAATGC
	R: CATAGGATCGCCACACGGATTAGC
ACTIN	F: ACCTGAGCGCAAGTACTCTGTCT
	R: CATCGTACTCCTGCTTGCTGAT

## Data Availability

Data presented in this study are available upon request from the corresponding author.

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
