# Peer review of "Effects of Insulin on Proliferation, Apoptosis, and Ferroptosis in Primordial Germ Cells via PI3K-AKT-mTOR Signaling Pathway"

_genes, 2023, doi:10.3390/genes14101975_

Round 1
Reviewer 1 Report
The authors described the effect of insulin on PGC proliferation in vitro, but their experiments lack novelty because it is already known that insulin is dispensable for PGC culture when other supplements are applied.
Q. PGC culture methods have been established for a long time as the authors mentioned, particularly the effect of insulin in PGC culture has known been dispensable when other supplements are provided (Whyte et al., 2015). Therefore, it is hard to accept the authors’ hypothesis of the paper.
Q. Please provide the proliferation rate of PGC when the insulin was removed.
Q. The authors did not include an insulin inhibitor group in their experiments.
Q. They did not describe the PGC culture media conditions in the M&M section.
Reviewer 2 Report
The authors analyzed the effects of insulin on proliferation, apoptosis and ferroptosis in Rugao yellow chicken embryonic PGCs. They found that insulin promotes PGCs proliferate and reduces apoptosis and ferroptosis in a dose-dependent manner and insulin activates PI3K/AKT/mTOR pathway. Overall, the results were clear and they are enough to support the conclusions. However, there are lots of mistakes and I’d like to comment on them.
And the main concern is that ref 30 and 31 already revealed that insulin is important for chicken PGC proliferation, so what is the main point of this manuscript? The appropriate concentration? The mechanisms? These points were not described in the discussion section. It seems that in this manuscript only the results were described clearly, but the future aspects and the comparison with previous studies were not described properly. So, this manuscript seems to be more technical reports.
L27 should change to “GSH/oxidized glutathione (GSSG)”
L53 should change to “stem cell factor (SCF)”
L54 -> “(LIF)”
L55 should change to “basic fibroblast growth factor (bFGF)”
L86 ). Fresh 4.5d
L95 note the concentration of trypsin solution
L96 200 ul -> 200 µl, and note what “PGC complete medium” is. (also in L99)
L105 note the catalogue and lot number, maker, and concentration of SSEA-1 antibody
L107 also the information of secondary antibody was not written
L108-110 please confirm the protocol. It seems strange.
L111 which hoechst used? (Catalogue number and maker)
L124 “Quantitative PCR” is fine
L125 the information is not enough, which kit was used to RNA extraction and cDNA synthesis. Also, how much of RNA was used for cDNA synthesis.
L131 should note not volume, but the concentration of primers
L139, L158 PGCs cells -> PGCs
L140-145 note the information of reagents (including antibody) used in this section.
L152 105/mL -> 10^5/mL
L158 106/mL -> 10^6/mL
L160 note information about Annexin V/PI
L169 SEEA-1 -> SSEA-1
L199 (Figure 1H-K) -> (Figure 2A-D)
L293-294 please re-write the sentence.
Fig. 1B In figure legend, it says “cultured without B-27”, but the explanation of Fig. 1B is “B-27 Minus insulin”, which is correct? If the latter is right, how to remove only insulin from B-27?
Fig. 1B and 1C, the picture is poor so the morphology of the cells are not clear (especially in Fig. 1B). Please use a different picture (Fig. 2A is OK). And the length of the white bar is not indicated.
Fig. 2B&2C too small to read.
Fig. 2D too faint, cannot see the signals
Reviewer 3 Report
Comments about the manuscript:
“Effects of insulin on proliferation, apoptosis, and ferroptosis in PGCs via PI3K-AKT-mTOR signaling pathway“
Primordial germ cells (PGCs) are essential for resource conservation and the recovery of endangered breeds in chickens. The aim of this study was to understand how certain factors act and how they are regulated. For this purpose, PGCs of Rugao yellow chicken were grown in media containing insulin at different doses. The authors were thus able to show that insulin intervenes in a dose-dependent manner in the expression of the mRNA of the genes involved, by improving cell proliferation and decreasing apoptosis and ferroptosis. They also showed that the PI3K-AKT-mTOR signaling pathway was involved
This work provides new knowledge and some improvements to the manuscript are needed.
Page 2, line 55. “Addition of LIF, SCF and bFGF inhibits apoptosis”: write also abbreviations in full when it is met for the first time in the text.
There is a lot of abbreviation: a lexical of these abbreviations at the end of the text would be useful.
Pages 2-3, lines 93-94. “separated in centrifugation machine”: give some details about centrifugation: speed in g number, duration.
Page 3, line 96. “centrifuged at 1000rpm”: use g number instead of rpm (rpm is dependent on the rotor length, so of the centrifuge model).
Page 3, line 103. “PBS”: For all the abbreviation, give the name in full when it is met for the first time in the text.
Page 3, line 11: write “Hoechst” instead of “hoechst” (with a first capital letter).
Page 3, lines 102-112. I don’t understand very well what is the technique described. The first part of paragraph: “Cells were collected in well plates, rinsed twice with PBS, fixed with 4% paraformaldehyde for 30 min, rinsed three times with PBS” concern cells;
the second part “0.1% Triton permeable membrane was treated for 15 min, rinsed three times with PBS, PBS with 10% FBS added was closed for 2 h against light and then a primary antibody SSEA-1 was added…” concerns a membrane (ELISA?).
This paragraph seems to be a mix of description of immunohistochemical or immunofluorescence reaction, and of an ELISA. The methods are not sufficiently described. Please, describe each method, one after other.
In a general manner, methods need to be described with more specificity.
Page 4, line 139. Write "RIPA" in full if it's not already encountered in the text.
Page 4, line 141. Write “2h” instead of “2hrs”.
Page 4, line 154. “absorbance at 532 nm and 412 nm was detected according to the instructions of MDA”: Briefly describe the method (like for other ones).
Page 5, figure 1: the scale bar of figure 2B is very small: indicate it in the legend of figure 2 or increase the scale.
Page 6, figure 2. give more explanations of the figure 2D. What are the numbers 0 - 100 on the pictures? mg/mL?
Page 6, line 199. “(Figure 1H-K),”: what are figures 1h-K? There are only figures 1A and B.
Page 7: figure 3 is not called in the text.
Page 10, lines 274-275. “expression levels of PI3K, Akt and mTOR genes”: use italics to write the gene names.
Page 11, lines 296-297. “In the present study, we found that removal of insulin from B-27 PGCs 296
significantly did not proliferate”: This sentence is difficult to understand. What doesn't proliferate? Cells? I suggest to write: "In the present study, we found that after insulin removal, B-27 PGCs have not proliferated significantly”.
Page 12, line 340. “previous results”: A reference would be helpful.
Round 2
Reviewer 1 Report
All inquiries have been resolved.